# Shade Avoidance and Light Foraging of a Clonal Woody Species, *Pachysandra terminalis*

**DOI:** 10.3390/plants10040809

**Published:** 2021-04-20

**Authors:** Risa Iwabe, Kohei Koyama, Riko Komamura

**Affiliations:** Laboratory of Plant Ecology, Department of Agro-Environmental Science, Obihiro University of Agriculture and Veterinary Medicine, Inadacho, Obihiro, Hokkaido 080-8555, Japan; s17120031@st.obihiro.ac.jp (R.I.); s17120080@st.obihiro.ac.jp (R.K.)

**Keywords:** clonal plants, clonal species, phalanx, guerrilla, light foraging, phenotypic plasticity, morphology, shade tolerance, shade acclimation, forest understory

## Abstract

(1) Background: A central subject in clonal plant ecology is to elucidate the mechanism by which clones forage resources in heterogeneous environments. Compared with studies conducted in laboratories or experimental gardens, studies on light foraging of forest woody clonal plants in their natural habitats are limited. (2) Methods: We investigated wild populations of an evergreen clonal understory shrub, Japanese pachysandra (*Pachysandra terminalis* Siebold & Zucc.), in two cool-temperate forests in Japan. (3) Results: Similar to the results of herbaceous clonal species, this species formed a dense stand in a relatively well-lit place, and a sparse stand in a shaded place. Higher specific rhizome length (i.e., length per unit mass) in shade resulted in lower ramet population density in shade. The individual leaf area, whole-ramet leaf area, or ramet height did not increase with increased light availability. The number of flower buds per flowering ramet increased as the canopy openness or population density increased. (4) Conclusions: Our results provide the first empirical evidence of shade avoidance and light foraging with morphological plasticity for a clonal woody species.

## 1. Introduction

Clonal plants produce their offspring via not only sexual reproduction but also asexual reproduction, in which new individual clones with an identical gene set, called ramets, are generated [1,2,3,4,5,6]. Elucidating the mechanism by which clones forage resources in heterogeneous environments is a central issue in clonal plant ecology [4,5,7,8,9,10,11,12,13,14,15,16,17,18,19,20,21,22], plant growth modeling [23], and their application in vegetation management [17,18,19,20,24,25,26,27,28,29,30,31]. Additionally, the spatial arrangement of ramets determines the reproductive success of clonal plants; this is because aggregation of ramets that belong to the same genets leads to an increased percentage of geitonogamous self-pollination [6].

Plants in forests are often shaded by neighboring plants or canopy trees [20,32,33,34,35,36,37,38,39]. The theory of resource foraging predicts that clonal plants selectively deploy ramets in places where resources are locally abundant [7,8,9,10,11,12,18,19,21,40,41]. The foraging of favorable resource-rich patches acts as a mechanism of escape from unfavorable patches, including shaded patches [9,12,18]. Avoidance of shade can be achieved at multiple levels [4,19,38]. In shade, clonal plants often increase specific rhizome length (i.e., length per unit mass) [7,18] and elongate their internodes [7,18,38,42] or increase their petiole length [38,42,43]. These increases in rhizome and petiole length are shade-induced spacer elongation phenomena [42,44]. To date, several researchers have investigated the resource foraging theory by experimentally studying clonal plants [7,8,9,11,19,38,41,44].

Most of the previous studies on resource foraging of clonal plants were performed using plants that were grown in pots or trays [7,8,9,18,19,24,40,42,44,45,46,47] and/or inside laboratories or greenhouses [7,8,19,24,45] rather than in their natural habitats. Nevertheless, it has been discussed that plants respond to their environment in different ways depending on whether they are grown in pots or in their natural habitats [5,48,49]. Plants in natural habitats experience greater fluctuations in their environment compared with plants that grow in pots or in greenhouses [32,48]. Particularly, artificial shading using shade cloths or films (e.g., [7,10,17,18,42,44,45,46,47,50,51,52,53,54]) does not produce sunflecks experienced by plants grown in natural forest understory [32,48]. However, compared with studies in laboratories or gardens, studies on resource foraging of wild plants in their natural habitats (e.g., [12,38,41]) are limited. Moreover, most of the previous studies on resource foraging of clonal plants have investigated herbaceous [7,8,9,10,11,12,16,17,18,19,20,24,38,40,41,42,44,46,47] or bamboo [13,15] species. The reproductive biology [1,29,55,56,57,58,59] and resultant genetic structure of populations [60,61,62] of clonal woody species have been studied intensively. However, to the best of our knowledge, no studies have quantitatively investigated light foraging or shade avoidance for clonal woody species. Hence, it is unclear whether the theory of light foraging can be applied to woody clonal plants in forest understory vegetation.

Japanese pachysandra (*Pachysandra terminalis* Siebold et Zucc.; Buxaceae) is a short-stature (ramet height approximately 15–35 cm at the study site), evergreen woody clonal species [63,64] that propagates using horizontal (plagiotropic) rhizomes (i.e., belowground stems) [65]. Each ramet comprises a single (usually unbranched) vertical aboveground stem and evergreen leaves with its appearance similar to an erect herb (Figure 1). This species forms a dense stand in a relatively well-lit place (Figure 1a) and a sparse stand in a shaded place (Figure 1b). It is a monoecious species with unisexual flowers, and the fruits are white drupes (Figure 1c). This species is distributed in East Asia [63], and it is widely used as a ground cover plant in Japan (pers. obs.) and several other countries [53,54,66,67,68]. Reports by Jeong and Kim [53] and Lee et al. [54] clarified mechanisms of shade acclimation at the level of an individual leaf or ramet (e.g., leaf photosynthetic traits and plasticity in aboveground vertical stem length, etc.), which were generally in agreement with typical shade acclimation responses of nonclonal plants. However, the mechanisms of light foraging and shade avoidance by the deployment of new ramets as a clonal plant were not investigated in these studies. Therefore, in the present study, we investigated a forest woody clonal plant that grows in its natural habitat to clarify the mechanism underlying light foraging and shade avoidance for this species.

## 2. Results

### 2.1. Ramet Population Density

Ramet population density increased as canopy openness increased (*p* < 0.01, Figure 2a). Consequently, the leaf area index (LAI, i.e., total leaf area per unit land area) increased as canopy openness increased (*p* < 0.01, Figure 2b). The positive correlation between the LAI and canopy openness was caused by the increment in correlated increase in ramet population density and canopy openness, rather than by the increment in individual leaf size or ramet size. The individual leaf area (*p* = 0.583), mean whole-ramet leaf area (*p* = 0.439), and mean ramet height (*p* = 0.434) did not significantly increase with the increasing canopy openness.

### 2.2. Morphological Plasticity

This species has two morphologically distinct types of horizontal rhizomes: thin and thick (Figure 3; see the Discussion for these two types). The mean specific rhizome length (SRL; i.e., length per unit mass) was significantly higher for thin rhizomes than for thick rhizomes (*p* < 0.001). For both thin and thick rhizomes, the SRL increased in shade (thin: *p* = 0.0356; thick: *p* = 0.0486). Consequently, the overall SRL, calculated by dividing the total rhizome length (i.e., sum of the total lengths of thick and thin rhizomes) by the total mass, increased in shade (*p* = 0.0241; Figure 4a). The increased SRL resulted in the lower population density in shade (*p* < 0.001; Figure 4b).

Although SRL decreased in shade for both types of rhizomes, responses of rhizome length per ramet to different light availability differed between thin and thick rhizomes. The rhizome length per ramet of thin rhizomes increased in shade, though the results were not significant (*p* = 0.0634; Figure 4c). The modest elongation of the thin rhizomes resulted in a lower ramet population density in shade (*p* < 0.001; Figure 4d). By contrast, the length of the thick rhizomes per ramet did not show clear dependency on canopy openness (*p* = 0.688; Figure 4e). In addition, contrary to our expectation, petiole length, which we thought would increase in shade, decreased in shade (*p* < 0.01; Figure 4f).

### 2.3. Sexual Reproduction

Probability of having at least one flower bud did not significantly increase as canopy openness increased (*p* = 0.959). Nevertheless, among the ramets with flower buds, the total number of flower buds on each ramet significantly increased as the canopy openness or ramet density increased (*p* < 0.01 for both cases, Figure 5a,b).

## 3. Discussion

Higher specific rhizome length (SRL) resulted in lower ramet density in shade (Figure 2a; Figure 4a,b). The SRL reflects the achievement of unit length of elongation per unit investment of biomass (similar to specific leaf area [32,69,70,71,72,73]). A higher SRL in shade was consistent with the results of other herbaceous clonal species (*Glechoma hederacea* [7] and *Reynoutria japonica* [18]). However, to the best of our knowledge, our result provides the first of its kind for a woody clonal species living in their natural forest habitats. Previous studies on herbaceous species have shown that the plasticity of horizontal stems differed among clonal species. For some species, the elongation of the stolon or rhizome internode length was observed (*Cymbalaria muralis* [42], *Fragaria vesca* [42], *Glechoma hederacea* [7], *Hydrocotyle bonariensis* [9], *H. vulgaris* [10], *Lamium galeobdolon* (syn. *Lamiastrum galeobdolon*) [10], *Potentilla anglica* [46], *Reynoutria japonica* [18], and *Trifolium repens* [44]). Conversely, the elongation was not observed for other species (*Anagallis tenella* [42], *Glechoma hirsuta* [42], *Potentilla reptans* [42,46], and *Ranunculus repens* [42]). *Pachysandra terminalis* has two types of horizontal rhizomes: thin rhizomes are milky-brown-white, whereas thick rhizomes are usually green or purple-red (Figure 3). Yoshie et al. [65] discussed that thin rhizomes sprout from existing underground rhizomes, whereas thick rhizomes originate from aboveground vertical stems that gradually descend and become underground horizontal rhizomes with roots. A thick rhizome might have an additional function as an underground storage organ, as a meter of each type allows this species to deploy offspring ramets one meter apart from the parent ramets.

The mean ramet leaf area did not increase as the canopy openness increased, consistent with the result observed for *G. hederacea* [7]. Given that neither ramet leaf area nor individual leaf area changed with the light availability, the increased population density was responsible for the increment of LAI (Figure 2b). Thus, although this species can tolerate shade, our results revealed that the effectiveness of ground cover as a gardening plant increased as the light availability increased. The number of flower buds per flowering ramet increased as canopy openness or ramet density increased (Figure 5). The increment in investment in sexual reproduction with the increment in ramet density is consistent with the results observed for dewberries (*Rubus* spp.) [1] and the clonal herb *Ranunculus reptans* [24].

We observed that the petiole length increased as the light availability increased (Figure 4e). Our result differs from most of the previous study results for the herbaceous clonal stoloniferous or rhizomatous species, where a longer petiole length in shade was observed (*Cymbalaria muralis* [42], *Fragaria vesca* [42], *Glechoma hederacea* [7,47], *G. hirsuta* [42], *Hydrocotyle vulgaris* [10], *Lamium galeobdolon* [10], *Potentilla anglica* [46], *P. reptans* [42,46], *Ranunculus repens* [42], *Trifolium fragiferum* [38,42], and *T. repens* [42,44]). However, no such trend was observed for *Petasites japonicus* [32]. Most of the herbaceous clonal species investigated so far have vertical petioles that directly elongate from horizontal aboveground stolons (but see the results related to an erect herb, *L. galeobdolon* [10]). Huber et al. [42,46] suggested a theory that vertically oriented (orthotropic) organs are more plastic in response to shade than horizontally oriented (plagiotropic) ones. Supporting this prediction, Huber et al. [42,46] showed that horizontal petioles that elongate from the vertical stems of erect herbaceous species were less plastic in response to shade compared with vertical petioles that elongate from horizontal stolons of clonal species from closely related taxa. Similar to these erect herbs, the species *P. terminalis* in the present study has horizontal petioles that elongate from vertical aboveground woody stems (which in turn connect to horizontal belowground rhizomes). The reason for this difference may be explained by the difference between vertical and horizontal structures of petioles, although other factors, such as the difference between herbaceous and woody species and/or between short-lived leaves of herbaceous species or long-lived evergreen leaves of woody species, may be present. In addition, as petioles do not only function as shade-avoidance structures but also as supporting tissues [74,75], their function could differ among the species with different leaf morphology. Currently, however, limited information is available on light foraging for woody clonal species.

Our study had several additional limitations. First, we did not investigate the special genetic structure of the populations. Therefore, although we found a difference in ramet density in different light environments, we cannot determine whether the ramets from the same genets aggregate in a dense stand (i.e., “phalanx form” [6,18,76,77]) or are sparsely dispersed and intermingled with other genets (i.e., “guerrilla form” [6,18,76,77]). The type of the spatial arrangement of ramets (phalanx or guerrilla) determines not only resource capture but also reproductive success, as an aggregation of ramets that belong to the same genets leads to an increased percentage of geitonogamous self-pollination [6]. Second, the total number of flowers may not provide an appropriate estimate of reproductive success through sexual reproduction because reproductive success is also determined by pollen limitation [78,79] and inbreeding depression caused by self-pollination [6,79,80]. Our data only showed that investment in sexual reproduction increased with the increment in ramet density, but it did not show the difference in reproductive success in different light environments. Third, only one woody clonal species was investigated for a short period. The differences among woody species from wider taxa and growth forms, especially shrub vs. tall trees and deciduous vs. evergreens, should be investigated in future studies. Given these important limitations, further studies are needed to reconfirm our findings before any generalization.

## 4. Materials and Methods

### 4.1. Study Sites

In 2020, we investigated wild populations in two nearby cool-temperate forests (approximately 3 km apart from each other) in an urban area of Obihiro City in Hokkaido, Japan. During the period in 1998–2017, the mean annual temperature and precipitation at the Obihiro Weather Station (approximately 6 km from the two study sites) were 7.2 °C and 937 mm, respectively [81]. The first site (F) was in the Forest of Obihiro (42°53′ N, 143°09′ E, altitude: 86 m a.s.l.), which is a secondary forest composed of a mixture of planted and regenerated trees comprising varieties of deciduous broad-leaved trees and evergreen coniferous trees. The second site (H) was located on the campus of the Hokkaido Obihiro Agricultural High School (45°52′ N 143°11′ E, altitude: 69 m a.s.l.). The forest comprises natural broad-leaved deciduous forests and planted coniferous forests, but the study was conducted only in a natural forest stand. The understory vegetation comprises a mixture of native species, which include *Pachysandra terminalis*, *Sasa chartacea* (Makino) Makino & Shibata, *Cardiocrinum cordatum* (Thunb.) Makino [78], and *Phryma esquirolii* H.Lév.

### 4.2. Sampling Strategy

We investigated 13 plots (F: *n* = 6, H: *n* = 7). In the Forest of Obihiro, *P. terminalis* covered the forest floor as discrete large patches, and one plot was established in each investigated patch. In the high school forest, the species continuously covered the range of ground of the investigated forest stand, and seven plots from different locations within the stand were established. Within each plot, we established two or three 1 m^2^ subplots, avoiding the edges of the patches or the stands. The total number of subplots was 33 (F: *n* = 14, H: *n* = 19). In one plot (H, #11), the field investigation was interrupted in summer due to a hornet nest in the plot; hence, some parameters (LAI and the ramet height) were not obtained.

### 4.3. Measurement of Canopy Openness

For each subplot, we estimated the canopy openness using hemispherical photographs [82,83,84] taken on cloudy days during the period of 4–6 May 2020, just after snow melt when the leaves of the upper canopy deciduous trees had not yet expanded. We used a light environment before canopy closure because photosynthesis before canopy closure is especially important for photosynthetic carbon gain for this understory shrub species [65,85]. The photographs were taken with a Nikon Fisheye Converter FC-E8 mounted on Nikon CoolPix P5100. The camera was set horizontally approximately 1 m above the ground using tubular spirit levels and a tripod. The images were binarized, and canopy openness was calculated using the CanopOn2 software [83].

### 4.4. Ramet Population Density in Spring and Sexual Reproduction

We counted the total number of ramets in each subplot from 29 April to 9 May, in 2020. Ramet population density in spring was defined as the number of ramets in each 1 m^2^ subplot. Each ramet was classified as either (1) a ramet with at least one flower bud or (2) a ramet without any flower buds, to calculate the percentage of flowering ramets in each subplot. The number of flower buds (i.e., the sum of female and male flower buds) was counted for all flowering ramets if the total number of flowering ramets in a plot did not exceed 10. If the number of flowering ramets in a plot exceeded 10, the number of flower buds was counted for the 10–15 haphazardly selected ramets.

### 4.5. Measurement of LAI

We estimated the leaf area on 13–14 July 2020, after the appearance of current-year leaves had completed. Five ramets that were adjacent to, and hence from the same stand of, each subplot were harvested. Immediately after sampling, the leaves were scanned with an A4 flatbed scanner (CanoScan LiDE 210; Canon, Tokyo). The individual leaf area was measured using ImageJ software [86], and the mean total leaf area per ramet was calculated. The leaf area index (LAI) was estimated as the product of the mean total leaf area per ramet and the number of ramets within each 1 m^2^ subplot. The height of ramets (defined as the distance between the ground surface and the point at which the highest leaf attached to the vertical stem) was measured for 8–12 ramets in each subplot on 30 July 2020.

### 4.6. Measurement of Rhizome Length and Ramet Population Density in Autumn

All belowground parts of *P. terminalis* were harvested in eight subplots (F: *n* = 6, H: *n* = 2) during 11–27 October 2020. The ramet population density in autumn was defined as the total number of ramets in each 1 m^2^ subplot. The two types of rhizomes (Figure 3) were sampled separately. Additionally, for each subplot, we sampled a total of 1 m of each type of rhizome, each of which comprised five to ten 10–20 cm parts. All of the harvested rhizomes were oven-dried at 70 °C for at least one week. The total dry mass of all rhizomes and 1 m samples for each subplot were measured with a precision balance. The total length of the rhizomes in each subplot was calculated by dividing the total mass by the mass per unit length of each type of rhizome. The total rhizome length was calculated as the sum of thick and thin rhizome lengths for each subplot. The mean rhizome length of each ramet was calculated by dividing the total rhizome length of each type by the number of ramets at the time of harvest. The specific rhizome length (SRL, i.e., length per unit mass) of each type of rhizome (thin and thick) was calculated as the inverse of the mass per unit length of the 1 m samples of each type. The overall SRL (thin and thick rhizomes combined) was calculated by dividing the total rhizome length by the total rhizome mass in each subplot; this value represents the proportions of thick and thin rhizomes and the SRL of each rhizome type.

The length of the petiole was measured for 15 leaves from five ramets (three leaves per ramet) in each subplot. Immediately after sampling, the leaves were scanned with an A4 flatbed scanner (CanoScan LiDE 210). The length of the petiole was measured using Image J software [86].

### 4.7. Statistical Analysis

All statistical analyses were performed with the statistical software R [87] using packages “cowplot” [88], “ggplot2” [89], and “lme4” [90]. Significance of the effect of each explanatory variable was tested with a generalized linear mixed model (GLMM) using the function *glmer* [90]. To predict binomial outcomes (a ramet either with at least one or without any flower bud), we used a logistic regression analysis with binomial error distribution (family = binomial (link = “logit”)) [91,92]. To predict positive and discrete dependent variables (number of ramets per unit land area and number of flower buds on each ramet), we used Poisson distribution (family = poisson (link = “log”)) [93,94]. To predict positive and continuous variables (the rest of the dependent variables), we used Gamma distribution (family = Gamma (link = “log”)) [95]. Following the arguments in [96], we constructed a model of maximal random effects structure by choosing all random effects that were justified by the design instead of choosing random variables via model selection. In the present case, random effects (i.e., ramet (only for petiole length and individual leaf area), subplot, plot, and site) were nested. When justified by the design, all these factors were included as nested random slopes and intercepts. The dataset presented in this paper is available as a part of the Appendix A.

## Figures and Tables

**Figure 1 plants-10-00809-f001:**
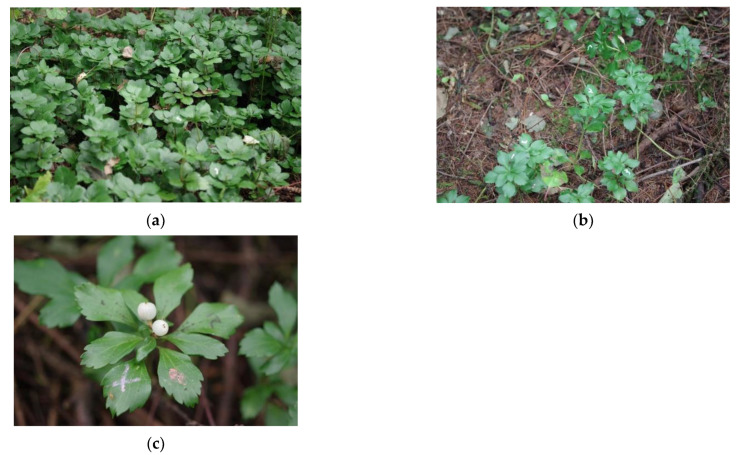
Photographs of *Pachysandra terminalis* Siebold et Zucc. (**a**) A dense stand in a relatively better-lit plot. (**b**) A sparse stand in a shaded plot. (**c**) Drupes. Markings or numberings by ink pens appear on some leaves. Photographs were taken on 1 October 2020 by Kohei Koyama. High-resolution images are available in the Appendix A.

**Figure 2 plants-10-00809-f002:**
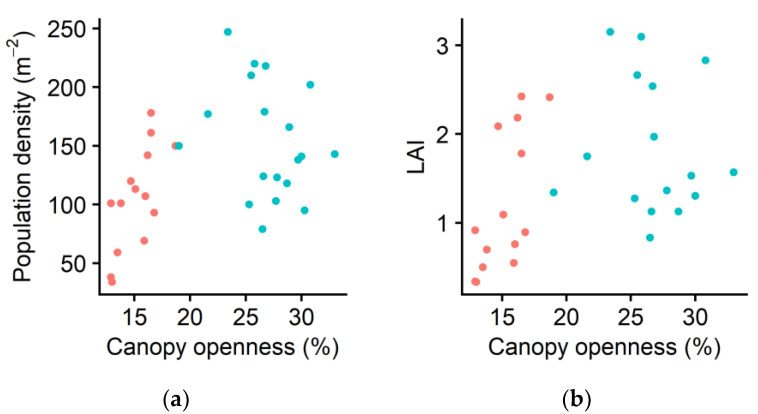
(**a**) Population density (i.e., the total number of ramets in each 1 m^2^ subplot) in spring and (**b**) leaf area index (LAI, i.e., total leaf area per unit land area) in relation to canopy openness in spring. Each symbol indicates the value of one subplot. Red: Forest of Obihiro. Blue: forest of the Hokkaido Obihiro Agricultural High School. The dataset is available as a part of the Appendix A.

**Figure 3 plants-10-00809-f003:**
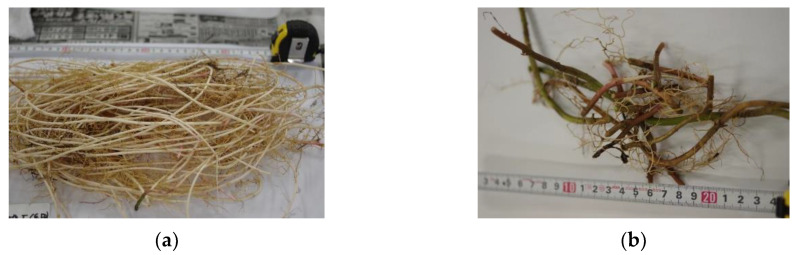
(**a**) Thin and (**b**) thick rhizomes. Photographs were taken on 18 October 2020, by Kohei Koyama. High-resolution images are available in the Appendix A.

**Figure 4 plants-10-00809-f004:**
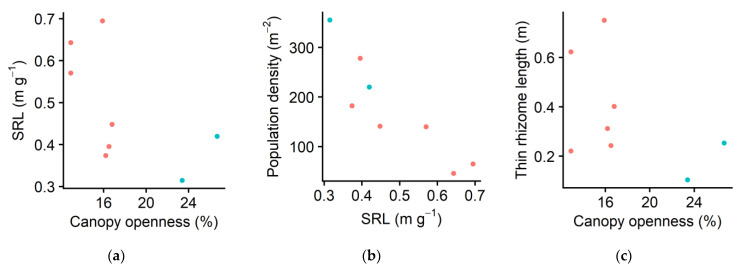
(**a**) Specific rhizome length (SRL; i.e., length per unit mass, thin and thick combined) in relation to canopy openness. (**b**) Population density (i.e., the number of ramets in each 1 m^2^ subplot in autumn) in relation to SRL. (**c**) Thin rhizome length per ramet (i.e., the total thin rhizome length in each subplot divided by the number of ramets in that subplot) in relation to canopy openness. (**d**) Dependence of population density on thin rhizome length per ramet. (**e**) Thick rhizome length per ramet in relation to canopy openness. (**f**) Mean petiole length in relation to canopy openness. Each symbol indicates the value of one subplot. Red: Forest of Obihiro. Blue: forest of the Hokkaido Obihiro Agricultural High School. The dataset is available as a part of the Appendix A.

**Figure 5 plants-10-00809-f005:**
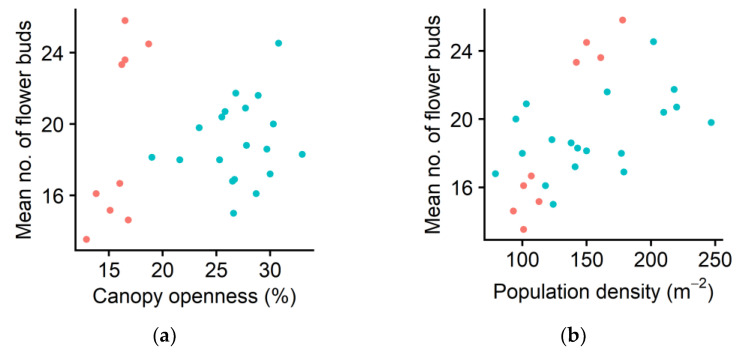
Mean number of flower buds on each ramet in relation to (**a**) canopy openness and to (**b**) population density (i.e., number of ramets in each subplot) in spring. Each symbol indicates the value of one subplot. Red: Forest of Obihiro. Blue: forest of the Hokkaido Obihiro Agricultural High School. The dataset is available as a part of the Appendix A.

## Data Availability

The datasets used in this article are available in the Appendix A.

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
