# Peer review of "Shade Avoidance and Light Foraging of a Clonal Woody Species, Pachysandra terminalis"

_plants, 2021, doi:10.3390/plants10040809_

Round 1
Reviewer 1 Report
Dear authors, here are my comments. L14-16, L.73-74: phalanx vs. guerilla is not necessarily about dense vs. sparse stands. Please read the manuscript "The Ecological and Evolutionary Consequences of Clonality for Plant Mating" by Valleyo-Marin et al. It is not possible (or at least it is hardly possible) to define which type of clonal architecture is present without genotyping. After reading the entire manuscript, here are my thoughts on this matter. Do not bother with the clonal architecture (guerilla vs. phalanx aspect) in the Abstract or the Discussion. Explain it shortly in the Introduction (following Valleyo-Marin et al.) and again mention it in the closing section. At the end of the Discussion (L276), please write something like "however, to obtain more reliable results which could elucidate this species clonal architecture in different locations, genotyping should be done“. As I already mentioned earlier, it is impossible to define which type of architecture prevails without genotyping. Nevertheless, this issue deserves to be mentioned. L117-118: Did you test the difference between thin and thick rhizome length per ramet? I apologize if I missed this, but if you didn’t test it, you can easily delete this sentence. Since both values (the increase of thin/thick rhizome length in shade) are not significant, this sentence can be misleading. L 173-179 and regarding the response on my earlier comment on sexual reproduction: there is no such thing as „...therefore, we believe the total number of flowers on each ramet is a suitable measure of reproductive allocation...“ in science. It is ok to believe in anything we want, but there have to be some pieces of evidence to back up these beliefs. Ok, moving on: if you believe that your approach is the right one, please explain it somewhere within the text, perhaps in the Discussion section (L247). It is not prohibited to enter the speculation zone, but then it should be nicely elaborated. To conclude: the manuscript is now improved, no doubt about it. However, it is obvious that there are some cavities in this research that can't be corrected at this stage. In such a situation, it is important to address these drawbacks and explain them as well as possible. This way there will be no misleading parts of the manuscript and the reader will be able to comprehend the most important presented information.Author Response
Author's Reply to the Review Report (Reviewer 1)
R1-1 (Reviewer 1’s 1st comment)
L14-16, L.73-74: phalanx vs. guerilla is not necessarily about dense vs. sparse stands. Please read the manuscript "The Ecological and Evolutionary Consequences of Clonality for Plant Mating" by Valleyo-Marin et al. It is not possible (or at least it is hardly possible) to define which type of clonal architecture is present without genotyping. After reading the entire manuscript, here are my thoughts on this matter. Do not bother with the clonal architecture (guerilla vs. phalanx aspect) in the Abstract or the Discussion. Explain it shortly in the Introduction (following Valleyo-Marin et al.) and again mention it in the closing section. At the end of the Discussion (L276), please write something like "however, to obtain more reliable results which could elucidate this species clonal architecture in different locations, genotyping should be done“. As I already mentioned earlier, it is impossible to define which type of architecture prevails without genotyping. Nevertheless, this issue deserves to be mentioned.
(Reply) Thank you for informing us of the relevant review paper (Vallejo-Marin et al. 2010 Ann Rev Ecol Evol Syst 41:193-213, doi:10.1146/annurev.ecolsys.110308.120258.). We have carefully read the paper and now understand that without genotyping, it is impossible to state whether the structure is “guerrilla” or “pharynx.”
(1) Per your suggestion, we removed the terms “guerrilla” and “pharynx” from the Abstract as well as from the Materials and Methods.
(2) Per your suggestion, we added the suggested limitations of the study at the end of the Discussion (lines 189–). Further, the paper (Vallejo-Marin et al., 2010) is now cited in the Introduction and Discussion.
R1-2
L117-118: Did you test the difference between thin and thick rhizome length per ramet? I apologize if I missed this, but if you didn’t test it, you can easily delete this sentence. Since both values (the increase of thin/thick rhizome length in shade) are not significant, this sentence can be misleading.
(Reply) Based on your question, we have revised the sentence and modified the descriptions in “2.2. Morphological Plasticity” and “4.6. Measurement of Rhizome Length” to clarify this point.
R1-3
L 173-179 and regarding the response on my earlier comment on sexual reproduction: there is no such thing as „...therefore, we believe the total number of flowers on each ramet is a suitable measure of reproductive allocation...“ in science. It is ok to believe in anything we want, but there have to be some pieces of evidence to back up these beliefs. Ok, moving on: if you believe that your approach is the right one, please explain it somewhere within the text, perhaps in the Discussion section (L247). It is not prohibited to enter the speculation zone, but then it should be nicely elaborated.
(Reply)
After reading the suggested paper (Vallejo-Marin et al., 2010), we agree with your statement that the total number of flowers alone is not sufficient to determine this clonal plant’s reproductive success. As sexual reproduction is not the focus of the present study, we added this limitation about sexual reproduction at the end of the Discussion (lines 196–) instead of providing a rationale for our argument about sexual reproduction.
R1-4
To conclude: the manuscript is now improved, no doubt about it. However, it is obvious that there are some cavities in this research that can't be corrected at this stage. In such a situation, it is important to address these drawbacks and explain them as well as possible. This way there will be no misleading parts of the manuscript and the reader will be able to comprehend the most important presented information.
(Reply)
Thank you for your constructive feedback. We revised the Discussion to include the limitations and caveats of our results. We believe that the revised manuscript contains sufficient information and arguments.

Reviewer 2 Report
The authors have provided qualified, meaningful responses to our suggestions contained in a previous review of this manuscript.
Author Response
Author's Reply to the Review Report (Reviewer 2)
Reviewer 2
The authors have provided qualified, meaningful responses to our suggestions contained in a previous review of this manuscript.
(Reply) Thank you very much for your encouraging feedback. We appreciate your time and careful review of our manuscript.

This manuscript is a resubmission of an earlier submission. The following is a list of the peer review reports and author responses from that submission.
Round 1
Reviewer 1 Report
Dear authors, I have thoroughly read your manuscript, and here are my comments. As you will see, my reading order was Discussion-Materials and methods-Results-Discussion, so I placed my comments accordingly.
In the Materials and methods section, while reading “Ramet Population Density in Spring and Sexual Reproduction” subsection (L211), one important issue crossed my mind. Throughout the manuscript, you are discussing the ramets density as a direct consequence of shade/ canopy openness. However, is it possible that the density is (also) linked to some other aspects of species biology, for instance, species clonal architecture or perhaps inter-genet competition? How many genets were present in the studied plots? What is the clonal architecture of the studied species (phalanx vs guerrilla)? What if, for instance, new seedlings more readily emerge in non-shade conditions, which consequently results in a greater density of both genets and ramets in sunny places? These are the question which deserves to be addressed. However, I am well aware that it is not possible to do that without genotyping, and I am not asking to do it. However, I think it is important to address these issues, perhaps in the Introduction or/and again in the Discussion section). Also, you should distance yourself from strong conclusions, not because I think they are incorrect, but because there are some other aspects that (possibly) can influence the ramet density/rhizome length.
L216: since the species is monoecious, I am not sure that it is correct to estimate sexual reproduction based on the total number of flowers. The ratio between male and female flowers can vary depending on different ecological factors, which can strongly influence the success of sexual reproduction. It would be more correct if the fruit set, and not the sum of all the flowers, was analyzed. This issue should be discussed in short. Once again, avoid strong conclusions.
L218-219: which selection criteria were followed when “the number of flower buds was counted for the 10–15 selected ramets”? How you managed, without any bias (in sense of the number of the flowers), to select these ramets?
L232: Why in only eight subplots were rhizomes harvested, and why in so uneven ratios (6:2), when in total, 33 subplots were established (14:19)? The obtained results for this part of the research should be discussed with great caution. Unfortunately, this part makes the central point of the research, but it is under-sampled! As I can see from the results section, some results lack statistical significance, which would possibly (or even likely) be obtained if the greater number of subplots were harvested.
L234: What is the story behind two types of rhizomes? Perhaps I missed it somewhere, but why these two types of rhizomes exist? Is their thickness related to their age? Are perhaps the thin ones from this vegetation season, and the thick ones are (substantially) older? In a sentence or two, this should be discussed, perhaps in Introduction or Discussion. EDIT: I see it now, in the Results section (L93). This part of the text should be moved to the Discussion section.
L102: The result is either significant or is not significant. p=0.06 is not significant, so please, correct this sentence. However, this is a real shame since this is supposed to be the most important result of the research.
L133: as already mentioned, no evidence of a longer thin rhizome in shade was obtained. The obtained result was not statistically significant. I am well aware that this result is THE main result of the research, but that does not change the fact.
To conclude: I like the idea behind the research and I think you wrote a good manuscript. However, I am sorry to see that the main result did not support the main hypothesis. I assume you had justified reasons why you reduced the number of subplots where rhizomes were harvested so drastically. I believe that the obtained result would be of significant value if the greater number of subplots remained. This way, no strong conclusion can be made. I am well aware that some other obtained results also point towards the same conclusion (although some results suggest the opposite, as well). However, the only result which directly characterizes the length of the rhizomes per ramet failed to support the hypothesis.
Reviewer 2 Report
Highly qualified good structured contribution.
Our comments (below) focused mostly on support of the authors. Some suggestions are listed too.
Results
Deviation of the relationship between the data of Fig. 2 is too large to give a confident conclusion about the presence or absence of correlation between them. This is also evident by statistics: p = 0.434 seems too low for your conclusions. Why did not you present corresponding statistical analytical dependencies and linear or other representations in the figures 2-5?
Discussion
Line 133
Longer thin rhizome length per ramet and higher specific rhizome length (SRL) in shade Figure 4a, d) resulted in lower ramet density in shade (Figure 2a; Figure 4b).
Yes. This statement is undoubtable. Would you add some novelty to it…
Line 145
…our study is the first to quantitatively report that shade-induced spacer elongation occurs in woody species living in their natural forest habitats.
Yes, indeed. This is a novel result.
Line 156
We observed that the petiole length increased as the light availability increased (Figure 4e). Our result differs from most of the previous study results for the herbaceous clonal stoloniferous or rhizomatous species, where a longer petiole length in shade was observed.
Yes, it is interesting. Please give your own explanation to this phenomenon. Do not to rely on other researcher’s point of view only.
Please give more attention to your statistical results.